# Effect of acute kidney injury on the patients with hepatocellular carcinoma undergoing transarterial chemoembolization

**Won Sohn**[ID]©, **Cheol Bae Ham**©, **Nam Hee Kim**, **Hong Joo Kim**[ID], **Yong Kyun Cho**, **Woo Kyu Jeon**, **Byung Ik Kim***

Division of Gastroenterology, Department of Internal Medicine, Kangbuk Samsung Hospital, Sungkyunkwan University School of Medicine, Seoul, South Korea

© These authors contributed equally to this work.
* bik.kim@samsung.com

**Data Availability Statement:** All relevant data are within the manuscript and its Supporting information file.

## Abstract

The purpose of this study was to investigate the effect of acute kidney injury (AKI) on the prognosis of patients with hepatocellular carcinoma (HCC) undergoing transarterial chemoembolization (TACE). A total of 347 HCC patients with Child-Pugh class A and pre-TACE serum creatinine (SCr) $\leq$1.5 mg/dL undergoing TACE as an initial therapy 2000–2014 were analyzed. Overall survival with related risk factors including AKI was investigated. We assessed AKI based on the International Club of Ascites (ICA)-AKI criteria. The mean age was 60.9 years. Of 347 patients, death was observed in 109 patients (31.4%). The mean SCr levels at pre-TACE, one day, two months, and four months after TACE were 0.9, 0.9, 0.9, and 1.1 mg/dL, respectively. The AKI within four months after TACE developed in 37 patients (11%). The AKI stages were non-AKI in 310 (89%), stage 1 in 10 (3%), stage 2 in 10 (3%), and stage 3 in 17 patients (5%). Multivariable analysis showed that the risk factors for overall survival were serum albumin $\leq$3.5 g/dL (hazard ratio [HR] 1.58, $p$ = 0.027), BCLC stage B (HR 2.07, $p$ = 0.008), BCLC stage C (HR 3.96, $p$<0.001), bilobar tumor location (HR 1.66, $p$ = 0.022), AKI stage 1 (HR 6.09, $p$<0.001), AKI stage 2 (HR 8.51, $p$<0.001), and AKI stage 3 (HR 17.64, $p$<0.001). AKI is a crucial prognostic factor for overall survival in HCC patients undergoing TACE. The assessment of AKI based on the ICA-AKI criteria can facilitate evaluation of the prognosis of HCC patients undergoing TACE.

## Introduction

Hepatocellular carcinoma (HCC) is one of the most common malignancies and its mortality rate is 8.2% of all cancer-related deaths globally [1]. The prognosis of HCC depends on the liver function of patients as well as the tumor burden. The Barcelona Clinic Liver Cancer (BCLC) staging system, considering liver function and tumor burden, has been widely used to determine the treatment modality or to assess the prognosis of the patients with HCC [2, 3]. HCC mostly develops on a background of chronic liver disease or liver cirrhosis [4]. Thus, the management of liver cirrhosis is crucial to improving the prognosis of patients with HCC.

**Funding:** The authors received no specific funding for this work.

**Competing interests:** The authors have declared that no competing interests exist.

Acute kidney injury (AKI) is considered a crucial factor for the prognosis of patients with live cirrhosis. AKI is especially associated with poor overall survival in cirrhosis [5, 6]. It sometimes develops hepatorenal syndrome type 1 which is a rapidly progressive renal failure and results in death [7]. The definition of AKI in liver cirrhosis has been update. The traditional concept of AKI in liver cirrhosis was that a rise of more than 50% in serum creatinine (SCr) and final SCr $\geq$1.5 mg/dL [8]. Recently, the International Club of Ascites (ICA) suggested an updated criteria of AKI (ICA-AKI) in liver cirrhosis which is a rise in SCr $\geq$0.3 mg/dl within 48 hours or an increase in SCr $\geq$50% compared to the baseline value [9, 10]. Also, the ICA-AKI criteria concretely classified AKI status into ICA-AKI stage 1–3. The ICA-AKI criteria predict in-hospital mortality of cirrhotic patients treated in intensive care units [11].

Transarterial chemoembolization (TACE) has been widely used to treat unresectable HCC. TACE is a treatment modality for intermediate stage of HCC since the HCC patients treated with TACE had the gain of overall survival in the randomized controlled clinical trials [12, 13]. The risk of AKI was higher in HCC patients who underwent TACE than in those did not [14]. However, there is little evidence if the updated AKI criteria predicted the prognosis of the HCC patients who underwent TACE. The purpose of this study was to investigate the effect of AKI on the prognosis of HCC patients undergoing TACE based on the ICA-AKI criteria.

## Methods

### Study design and patients

This study was conducted based on a historical cohort of Kangbuk Samsung Hospital, Seoul, Korea [15]. A total of 547 HCC patients underwent TACE as an initial treatment for HCC January 2000 –December 2014. We excluded 200 patients with the following reasons: (i) Child-Pugh class B or C (n = 184); (ii) no information of serum creatinine (SCr) level during follow-up (n = 10); (iii) follow-up loss within 1 month (n = 2); (iv) pre-TACE SCr >1.5 mg/dL (n = 4). Finally, a total of 347 HCC patients with Child-Pugh class A and pre-TACE SCr $\leq$1.5 mg/dL undergoing TACE as an initial therapy were analyzed. HCC was diagnosed based on the American Association for the Study of Liver Diseases (AASLD) guideline [3]. This study was conducted in the retrospective manner using a historical cohort. Therefore, the requirement for informed consent was waived. All data were fully anonymized before the authors accessed them and/or whether the institutional review board (IRB) waived the requirement for informed consent. The study was approved by the IRB of Kangbuk Samsung Hospital (IRB No. 2015-08-042).

### Transarterial chemoembolization

The hepatic artery was catheterized after celiac and superior mesenteric arteriography. The tumor-feeding arteries were also catheterized for a selective embolization to treat HCC. After the selection of the feeding artery to HCC, the segmental embolization of the tumor supplying artery was performed using an emulsion of iodized oil (Lipiodol; Andre Gurbet, Aulnay-sous-Bois, France) with doxorubicin or cisplatin. The dose of emulsion agent was determined pursuant to the tumor size, number of tumors, supplying vessel, and underlying liver function. This emulsion agent was infused until stasis of arterial flow had been achieved and/or the iodized oil was observed in the portal branches. Thereafter, embolization with gelatin sponge particles (1–2 mm in diameter; Gelfoam; Upjohn, Kalamazoo, MI) was performed. Immediately after embolization, an angiography was conducted again to evaluate the extent of the vessel occlusion.

## Assessment of outcome and follow-up

The patients were evaluated for age, sex, etiology of liver disease, platelet count, prothrombin time-international normalized ratio (PT-INR), serum albumin level, total bilirubin, aspartate transaminase (AST), alanine transaminase (ALT), creatinine, serum alpha-fetoprotein (AFP), model for end-stage liver disease (MELD) score, presence of ascites, presence of esophageal or gastric varices, maximal tumor size, tumor number, tumor location, portal vein invasion, Barcelona Clinic Liver Cancer (BCLC) stage, chemoembolization agents, diabetes mellitus, hypertension, and use of medication (metformin, beta-blocker, diuretics and angiotensin converting enzyme (ACE) inhibitor / angiotensin receptor blocker (ARB)). Overall survival after TACE was investigated in the enrolled patients. Overall survival was defined as the time from the date of the first TACE therapy to death.

The AKI was evaluated based on ICA-AKI criteria.[10] It was defined as an increase in serum Cr ≥0.3 mg/dl (≥26.5 μmol/L) within 48 hours or a percentage increase in serum Cr ≥50% from baseline which was known, or presumed, to have occurred within the prior seven days. Baseline serum Cr was defined as the value obtained within the previous three months. When the patients did not have a value of Cr within the previous three months, we used the value closest to the admission day to the hospital. If the patients had not been checked with serum Cr before admission, the baseline Cr was determined as the value on admission with cirrhosis to be. We assessed the AKI stage using SCr at one day, two months, and four months after the TACE.

The AKI stage after the TACE is defined as follow: non-AKI, no increase of SCr or increase of SCr <0.3 mg/dl; stage 1, increase of SCr ≥0.3 mg/dl or SCr ≥1.5—two times from baseline; stage 2, increase of SCr ≥2—three times from baseline; stage 3, increase of SCr ≥three times from baseline, or SCr ≥4.0 mg/dl with an acute increase of ≥0.3 mg/dl, or initiation of renal replacement therapy.

## Statistical analysis

Baseline characteristics were analyzed using descriptive statistics. The mean value with standard deviation (SD) was used in the continuous variables while frequencies with percentages were used in the categorical variables. The univariable and multivariable logistic regression models were applied to assess the risk factors for AKI development after TACE. The Kaplan-Meier curve was used to calculate overall survival after TACE. The log-rank test was applied to compare overall survival according to each risk factor. The univariable and multivariable Cox regression models were conducted to investigate the risk factors for overall survival after TACE. Multivariable analysis in logistic and Cox regression models was done in the following manners. First, we included in the multivariable analysis any variable that was associated with the outcome at a P-value of <0.10, regardless of whether or not the variable was associated with the risk factor. Second, for multivariable regression analysis, the subjects of analysis should have at least ten outcomes for each independent variable in the multivariable model. Third, multivariable analysis was performed using a forward conditional stepwise procedure to avoid multicollinearity. The statistical analyses were performed using SPSS version 18.0 (SPSS Inc, Chicago, Ill). A statistical significance was considered as positive if the *p*-value was <0.05 in a two-sided test.

## Results

### Baseline characteristics

Baseline characteristics of enrolled patients are presented in Table 1. A total of 347 HCC patients undergoing TACE was analyzed. The mean age of the patients was 60.9 and the

**Table 1. Baseline characteristics of the enrolled patients (N = 347).**

| | Number / mean ± S.D. |
|---|---|
| Age | 60.9 ± 10.6 |
| Male / female (N, %) | 277/70 (80/20) |
| Etiology of liver disease (N, %) | |
| Chronic hepatitis B | 260 (75) |
| Chronic hepatitis C | 36 (10) |
| Alcoholic liver disease | 41 (12) |
| Others | 10 (3) |
| Platelet count (x$10^3$/mm$^2$) | 158.8 ± 81.3 |
| Prothrombin time (INR) | 1.1 ± 0.1 |
| Albumin (g/dL) | 3.8 ± 0.4 |
| Total bilirubin (mg/dL) | 1.0 ± 0.4 |
| AST (U/L) | 74.7 ± 110.4 |
| ALT (U/L) | 51.3 ± 67.0 |
| Creatinine (mg/dL) | 0.9 ± 0.3 |
| Serum AFP level (median with interquartile range, ng/mL) | 23.3 (4.3–271.1) |
| MELD score | 8.4 ± 1.9 |
| Presence of ascites (N, %) | 60 (17) |
| Presence of varices (N, %) | 138 (40) |
| Maximum diameter of tumor (cm) | 5.3 ± 4.3 |
| Tumor number | |
| 1 | 170 (49) |
| 2 | 76 (22) |
| ≥3 | 101 (29) |
| Tumor location | |
| Unilobar | 238 (69) |
| Bilobar | 109 (31) |
| Portal vein invasion (N. %) | 87 (25) |
| Tumor stage (BCLC stage) | |
| Very early (stage 0) | 23 (6) |
| Early (stage A) | 124 (36) |
| Intermediate (stage B) | 93 (27) |
| Advanced (stage C) | 107 (31) |
| Chemoembolization agent (N, %) | |
| Doxorubicin | 284 (82) |
| Cisplatin | 63 (18) |
| Diabetes mellitus (N, %) | 91 (26) |
| Hypertension (N, %) | 118 (34) |
| Use of metformin (N, %) | 39 (11) |
| Use of beta-blocker (N, %) | 64 (18) |
| Use of diuretics (N, %) | 49 (14) |
| Use of ACE inhibitor / ARB (N, %) | 64 (18) |

*Abbreviation: S.D., standard deviation; INR, international normalized ratio; AST, aspartate aminotransferase; ALT, alanine aminotransferase; MELD, model for end-stage liver disease; BCLC, Barcellona Clinic Liver Cancer; ACE, angiotensin converting enzyme; ARB, angiotensin receptor blocker.

majority of the patients were male (80%). The main etiologies of liver disease were chronic hepatitis B (n = 260, 75%), alcoholic liver disease (n = 41, 12%), and chronic hepatitis C (n = 36, 10%). The mean levels of PT-INR, albumin, total bilirubin, AST, ALT, and pre-TACE serum Cr were 1.1, 3.8 g/dL, 1.0 mg/dL, 74.7 U/L, 51.3 U/L and 0.9 mg/dL, respectively. The median level of serum AFP was 23.3 ng/mL (interquartile range: 4.3–271.1 ng/mL). The mean value of the MELD score was 8.4. The mean diameter of the maxima tumor was 5.3 ± 4.3 cm. The number of tumors was one in 170 patients (49%), two in 76 patients (22%), and more than three in 101 patients (29%). The location of tumors was unilobar in 238 patients (69%), and bilobar in 109 patients (31%). Portal vein invasion was present in 87 patients (25%). BCLC stages were stage 0 in 23 patients (6%), stage A in 124 patients (36%), stage B in 93 patients (27%), and stage C in 107 patients (31%). Chemoembolization agents were used with doxorubicin in 284 patients (82%) and cisplatin in 63 patients (18%). The total number of TACE sessions during the follow-up was one in 114 patients, two in 86 patients, three in 51 patients, four in 29 patients, and five or more in 67 patients (median: two sessions, range 1–12).

## AKI and related risk factors during TACE

AKI in HCC patients during TACE was demonstrated in Table 2. The mean level of serum Cr at pre-TACE was 0.9 ± 0.2 mg/dL. The total number of TACE sessions within four months was one in 224 patients (65%), two in 87 patients (25%), and three in 36 patients (10%). We evaluated the AKI stage at one day, two months, and four months after TACE. At one day after TACE, the mean level of serum Cr was 0.9 ± 0.2 mg/dL. AKI at one day after TACE occurred in seven patients (2%): non-AKI in 340 patients (98%), stage 1 in seven patients (2%), stage 2 in 0 patient (0%) and stage 3 in 0 patient (0%). At two months after TACE, the mean level of serum Cr was 0.9 ± 0.3 mg/dL. AKI at two months after TACE occurred in six patients (2%): non-AKI in 330 patients (97%), stage 1 in one patient (1%), stage 2 in two patients (1%) and stage 3 in three patients (1%). At four months after TACE, the mean level of serum Cr was 1.1 ± 0.8 mg/dL. AKI at four months after TACE occurred in 27 patients (8%): non-AKI in 292 patients (91%), stage 1 in five patients (2%), stage 2 in eight patients (3%) and stage 3 in 14 patients (4%). AKI within four months after TACE was also evaluated. Within four months after TACE, AKI developed in 37 patients (11%): non-AKI in 310 patients (89%), stage 1 in 10 patients (3%),

**Table 2. Acute kidney injury in patients with HCC who underwent TACE.**

| Mean ± S.D. / Number | Pre-TACE | 1 day after TACE (n = 347) | 2 months after TACE (n = 336) | 4 months after TACE (n = 319) | Within 4 months after TACE (n = 347) |
|---|---|---|---|---|---|
| Serum creatinine level (mg/dL) | 0.9 ± 0.2 | 0.9 ± 0.2 | 0.9 ± 0.3 | 1.1 ± 0.8 | |
| Development of acute kidney injury (n, %) [a] | | 7 (2%) | 6 (2%) | 27 (8%) | 37 (11%) |
| Stage of acute kidney injury (n, %) [a] | | | | | |
| Non-AKI | | 340 (98%) | 330 (97%) | 292 (91%) | 310 (89%) |
| Stage 1 | | 7 (2%) | 1 (1%) | 5 (2%) | 10 (3%) |
| Stage 2 | | 0 (0%) | 2 (1%) | 8 (3%) | 10 (3%) |
| Stage 3 | | 0 (0%) | 3 (1%) | 14 (4%) | 17 (5%) |

[a]stage 0, no increase of serum creatinine (Cr) or increase of serum Cr <0.3 mg/dl; stage 1, increase of serum Cr ≥0.3 mg/dl or serum Cr ≥1.5–2 times from baseline; stage 2, increase of serum Cr ≥2–3 times from baseline; stage 3, increase of serum Cr ≥3 times from baseline, or serum Cr ≥4.0 mg/dl with an acute increase of ≥0.3 mg/dl, or initiation of renal replacement therapy.

*Abbreviation: S.D., standard deviation; HCC, hepatocellular carcinoma; TACE, transarterial chemoembolization; AKI, acute kidney injury.

stage 2 in 10 patients (3%) and stage 3 in 17 patients (5%). Possible causes for AKI development after TACE were analyzed. Of 37 patients with AKI development, 8 patients had experienced increasing ascites. Use of ACE inhibitor or ARB, and beta-blocker was observed in 6 patients, and 6 patients, respectively. Bacterial infection after TACE was observed in 5 patients (spontaneous bacterial peritonitis in 4 patients; pneumonia in 1 patient).

Univariable and multivariable analyses of risk factors for AKI development within four months after TACE in HCC patients are demonstrated in Table 3. The univariable analysis showed that AKI development after TACE is associated with platelet count $\leq$100 x$10^3$/mm$^2$ (odds ratio [OR] 0.24 with 95% confidence interval (CI): 0.07–0.79, $p = 0.020$), serum AFP $\geq$100 ng/mL (OR 3.45 with 95% CI: 1.70–6.99, $p = 0.001$), BCLC stage C (OR 15.91 with 95% CI: 6.39–39.59, $p<0.001$), use of cisplatin as chemoembolization agents (OR 4.88 with 95% CI: 2.38–10.01, $p<0.001$), presence of ascites (OR 12.12 with 95% CI: 5.74–25.61, $p<0.001$) and sessions of TACE $\geq$2 (OR 2.09 with 95% CI: 1.05–4.15, $p = 0.035$). The multivariable analysis revealed that the risk factors for AKI development after TACE were BCLC stage C (OR 10.52 with 95% CI: 4.08–27.16, $p<0.001$) and presence of ascites (OR 7.13 with 95% CI: 3.17–16.04, $p<0.001$).

## Risk factors for overall survival after TACE

Of 347 patients, death occurred in 109 patients (31.4%). Overall survival rates at 24, 48, 72, 96 and 120 months were 74.9%, 64.1%, 51.9%, 48.4% and 45.2%, respectively (Fig 1). Univariable and multivariable analyses of risk factors for overall survival in HCC patients undergoing TACE are presented in Table 4. Univariable analysis showed that overall survival was associated

**Table 3. Risk factors for acute kidney injury in patients with HCC who underwent TACE [a].**

| | univariable OR (95% CI) | *p*-value | multivariable OR (95% CI) | *p*-value |
|---|---|---|---|---|
| Age (years)$\geq$60 | 0.73 (0.37–1.45) | 0.731 | | |
| Men | 2.23 (0.76–6.52) | 0.142 | | |
| Platelet count $\leq$100 x$10^3$/mm$^2$ | 0.24 (0.07–0.79) | 0.020 | | |
| Serum albumin $\leq$3.5 g/dL | 1.96 (0.97–3.96) | 0.061 | | |
| Total bilirubin $\geq$1.2 mg/dL | 1.83 (0.91–3.70) | 0.090 | | |
| Pre-TACE SCr $\geq$1.0 mg/dL | 0.91 (0.44–1.88) | 0.801 | | |
| Serum AFP $\geq$100 ng/mL | 3.45 (1.70–6.99) | 0.001 | | |
| MELD $\geq$10 | 1.08 (0.49–2.40) | 0.846 | | |
| BCLC stage C (vs. stage 0, A and B) | 15.91 (6.39–39.59) | <0.001 | 10.52 (4.08–27.16) | <0.001 |
| Chemoembolization agent: cisplatin (vs. doxorubicin) | 4.88 (2.38–10.01) | <0.001 | | |
| Ascites | 12.12 (5.74–25.61) | <0.001 | 7.13 (3.17–16.04) | <0.001 |
| Varices | 0.80 (0.39–1.63) | 0.543 | | |
| Number of TACE $\geq$2 sessions | 2.09 (1.05–4.15) | 0.035 | | |
| Diabetes mellitus | 1.05 (0.49–2.26) | 0.907 | | |
| Hypertension | 1.06 (0.52–2.16) | 0.878 | | |
| Use of metformin | 0.67 (0.20–2.30) | 0.526 | | |
| Use of beta-blocker | 1.25 (0.54–2.88) | 0.599 | | |
| Use of diuretics | 0.51 (0.15–1.72) | 0.275 | | |
| Use of ACE inhibitor / ARB | 1.25 (0.54–2.88) | 0.599 | | |

[a]acute kidney injury was evaluated within four months after TACE.

*Abbreviation: HCC, hepatocellular carcinoma; TACE, transarterial chemoembolization; OR, odds ratio; CI, confidence interval; SCr, serum creatinine; AFP, alpha-fetoprotein; MELD, The Model for End-stage Liver Disease; BCLC, Barcellona Clinic Liver Cancer; ACE, angiotensin converting enzyme; ARB, angiotensin receptor blocker.

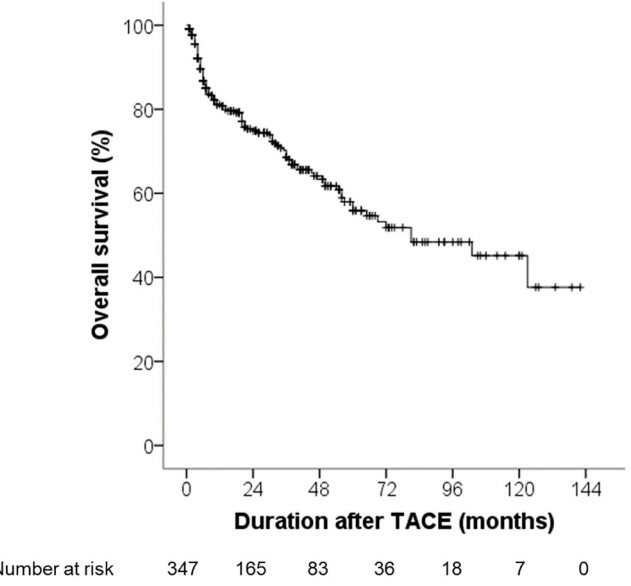

**Fig 1. Overall survival of the enrolled patients.**

with platelet count $\leq$100 x10³/mm² (hazard ratio [HR] 0.56 with 95% CI: 0.35–0.91, $p$ = 0.020), serum albumin $\leq$3.5 g/dL (HR 1.70 with 95% CI: 1.14–2.51, $p$ = 0.008), serum AFP $\geq$100 ng/ mL (HR 2.27 with 95% CI: 1.55–3.31, $p$ = 0.001), maximal tumor size $\geq$5 cm (HR 3.16 with 95% CI: 2.16–4.64, $p$<0.001), number of tumor $\geq$3 (HR 2.45 with 95% CI: 1.70–3.51, $p$<0.001), bilobar location (HR 2.47 with 95% CI: 1.69–3.63, $p$<0.001), portal vein invasion (HR 3.98 with 95% CI: 2.69–5.88, $p$<0.001), BCLC stage B (HR 2.24 with 95% CI: 1.32–3.81, $p$ = 0.003), BCLC stage C (HR 6.40 with 95% CI: 3.92–10.48, $p$<0.001), use of cisplatin as che-moembolization agents (HR 3.50 with 95% CI: 2.30–5.33, $p$<0.001), presence of ascites (HR 3.31 with 95% CI: 2.16–5.08, $p$<0.001), AKI stage 1 (HR 16.42 with 95% CI: 7.19–37.49, $p$<0.001), AKI stage 2 (HR 25.24 with 95% CI: 11.55–55.17, $p$<0.001) and AKI stage 3 (HR 34.52 with 95% CI: 16.52–72.14, $p$<0.001). Multivariable analysis indicated that the risk factors for overall survival were serum albumin $\leq$3.5 g/dL (HR 1.58 with 95% CI: 1.05–2.38, $p$ = 0.027), BCLC stage B (HR 2.07 with 95% CI: 1.21–3.54, $p$ = 0.008), BCLC stage C (HR 3.96 with 95% CI: 2.28–6.87, $p$<0.001), bilobar location of HCC (HR 1.66 with 95% CI: 1.08–2.55, $p$ = 0.022), AKI stage 1 (HR 6.09 with 95% CI: 3.68–19.67, $p$<0.001), AKI stage 2 (HR 8.51 with 95% CI: 2.54–14.57, $p$<0.001) and AKI stage 3 (HR 17.64 with 95% CI: 8.23–37.82, $p$<0.001).

The differences in overall survival of HCC patients undergoing TACE according to each risk factor are shown in Fig 2. Overall survival was significantly different according to serum albumin level ($p$ = 0.007). The 24-month, 48-month, 72-month, 96-month, and 120-month survival rates of patients were 76.9%, 69.1%, 59.7%, 54.9%, and 54.9%, respectively, in patients with serum albumin level >3.5 g/dL; and 69.2%, 50.1%, 31.8%, 31.8%, and 21.2%, respectively, in patients with serum albumin level $\leq$3.5 g/dL (Fig 2A). Overall survival was significantly different according to location of HCC ($p$<0.001). The 24-month, 48-month, 72-month, 96-month, and 120-month survival rates of patients were 82.9%, 71.9%, 59.8%, 57.4%, and 53.0%, respectively, in patients with unilobar involvement; and 56.2%, 45.4%, 31.3%, 25.0%, and 25.0%, respectively, in patients with biolobar involvement (Fig 2B). Over-all survival was significantly different according to BCLC stage ($p$<0.001). The 24-month, 48-month, 72-month, 96-month, and 120-month survival rates of patients were 90.3%,

**Table 4. Risk factors for overall survival in patients with HCC who underwent TACE.**

|  | univariable HR (95% CI) | p-value | multivariable HR (95% CI) | p-value |
|---|---|---|---|---|
| Age (years) ≥60 | 0.72 (0.49–1.04) | 0.081 |  |  |
| Men | 1.41 (0.86–2.31) | 0.178 |  |  |
| Platelet count ≤100 x $10^3$/mm$^2$ | 0.56 (0.35–0.91) | 0.020 |  |  |
| Serum albumin ≤3.5 g/dL | 1.70 (1.14–2.51) | 0.008 | 1.58 (1.05–2.38) | 0.027 |
| Total bilirubin ≥1.2 mg/dL | 1.35 (0.89–2.03) | 0.156 |  |  |
| Pre-TACE SCr ≥1.0 mg/dL | 0.74 (0.49–1.12) | 0.155 |  |  |
| Serum AFP ≥100 ng/mL | 2.27 (1.55–3.31) | <0.001 |  |  |
| MELD ≥10 | 1.00 (0.63–1.58) | 0.997 |  |  |
| BCLC stage B (vs. stage 0 & A) | 2.24 (1.32–3.81) | 0.003 | 2.07 (1.21–3.54) | 0.008 |
| BCLC stage C (vs. stage 0 & A) | 6.40 (3.92–10.48) | <0.001 | 3.96 (2.28–6.87) | <0.001 |
| Maximal tumor size ≥5 cm | 3.16 (2.16–4.64) | <0.001 |  |  |
| Number of tumor ≥3 | 2.45 (1.70–3.51) | <0.001 |  |  |
| Location of tumor: bilobar | 2.47 (1.69–3.63) | <0.001 | 1.66 (1.08–2.55) | 0.022 |
| Portal vein invasion | 3.98 (2.69–5.88) | <0.001 |  |  |
| Chemoembolization agent: cisplatin (vs. doxorubicin) | 3.50 (2.30–5.33) | <0.001 |  |  |
| Ascites | 3.31 (2.16–5.08) | <0.001 |  |  |
| Varices | 1.34 (0.92–1.95) | 0.132 |  |  |
| Diabetes mellitus | 1.05 (0.69–1.61) | 0.817 |  |  |
| Hypertension | 0.85 (0.57–1.28) | 0.434 |  |  |
| Use of metformin | 0.79 (0.41–1.51) | 0.468 |  |  |
| Use of beta-blocker | 0.76 (0.45–1.28) | 0.298 |  |  |
| Use of diuretics | 0.71 (0.40–1.27) | 0.243 |  |  |
| Use of ACE inhibitor / ARB | 0.84 (0.50–1.40) | 0.498 |  |  |
| AKI Stage 1 (vs. non-AKI) | 16.42 (7.19–37.49) | <0.001 | 6.09 (2.54–14.57) | <0.001 |
| AKI Stage 2 (vs. non-AKI) | 25.24 (11.55–55.17) | <0.001 | 8.51 (3.68–19.67) | <0.001 |
| AKI Stage 3 (vs. non-AKI) | 34.52 (16.52–72.14) | <0.001 | 17.64 (8.23–37.82) | <0.001 |

*Abbreviation: HCC, hepatocellular carcinoma; TACE, transarterial chemoembolization; HR, hazard ratio; CI, confidence interval; SCr, serum creatinine; AFP, alpha-fetoprotein; MELD, The Model for End-stage Liver Disease; BCLC, Barcellona Clinic Liver Cancer; ACE, angiotensin converting enzyme; ARB, angiotensin receptor blocker.

76.9%, 71.0%, 66.8%, and 66.8%, respectively, in patients with BCLC stage 0 or A; and 80.7%, 66.4%, 37.6%, 37.6%, and 30.1%, respectively, in patients with BCLC stage B; and 37.7%, 31.8%, 23.9%, 17.9%, and 17.9%, respectively, in patients with BCLC stage C (Fig 2C). Overall survival was significantly different according to AKI stage (*p*<0.001). The 12-month, 24-month, 48-month, 72-month, 96-month, and 120-month survival rates of patients were 89.1%, 82.7%, 70.8%, 57.3%, 53.4%, and 49.9%, respectively, in patients with non-AKI; and 18.8%, 0%, 0%, 0%, 0%, and 0%, respectively, in patients with AKI stage 1; and 0%, 0%, 0%, 0%, 0%, and 0%, respectively, in patients with AKI stage 2 or 3 (Fig 2D).

We performed subgroup analysis for the effect of AKI on overall survival according to BCLC stage, serum albumin level and presence of ascites. First, we evaluated the effect of AKI stage on overall survival of HCC patients undergoing TACE according to BCLC stage 0/A vs. BCLC stage B/C. In a subgroup of BCLC stage 0/A, overall survival was significantly different according to AKI stage (*p*<0.001). The 12-month, 24-month, 48-month, 72-month, 96-month, and 120-month survival rates of patients were 99.3%, 92.5%, 80.8%, 75.6%, 72.0%, and 72.0%, respectively, in patients with non-AKI; and 0%, 0%, 0%, 0%, 0%, and 0%, respectively, in patients with AKI stage 2 or 3. In a subgroup of BCLC stage B/C, overall survival was

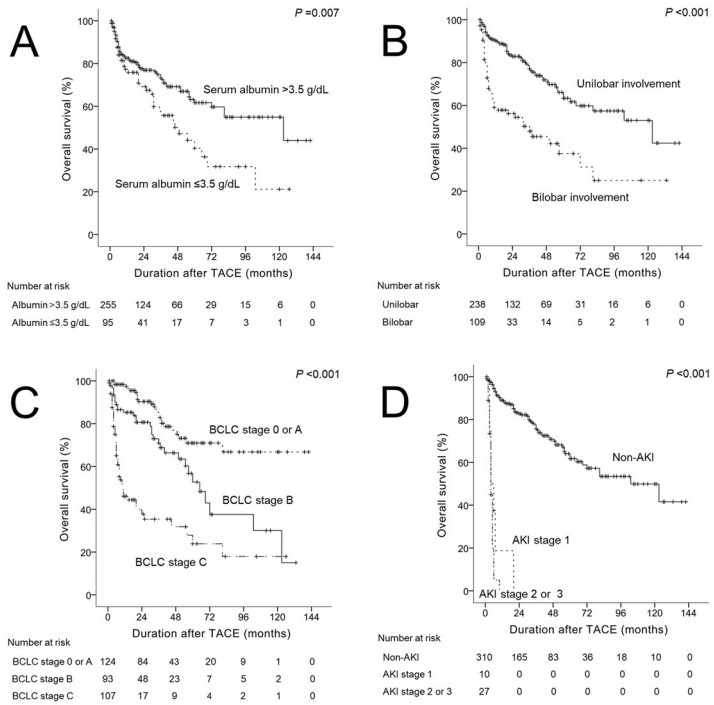

**Fig 2. Overall survival according to risk factors in patients with HCC who underwent TACE: (A) serum albumin level, (B) tumor location: unilobar vs bilobar, (C) BCLC stage, and (D) AKI stage.** *Abbreviation: HCC, hepatocellular carcinoma; TACE, transarterial chemoembolization; BCLC, Barcellona Clinic Liver Cancer; AKI, acute kidney injury.

significantly different according to AKI stage ($p<0.001$). The 12-month, 24-month, 48-month, 72-month, 96-month, and 120-month survival rates of patients were 78.8%, 72.7%, 60.2%, 75.6%, 37.2%, and 28.3%, respectively, in patients with non-AKI; and 18.8%, 0%, 0%, 0%, 0%, and 0%, respectively, in patients with AKI stage 1; and 0%, 0%, 0%, 0%, 0%, and 0%, respectively, in patients with AKI stage 2 or 3. Second, we evaluated the effect of AKI stage on overall survival of HCC patients undergoing TACE according to serum albumin ≤3.5 g/dL vs. >3.5 g/dL. In a subgroup of serum albumin >3.5 g/dL, overall survival was significantly different according to AKI stage ($p<0.001$). The 12-month, 24-month, 48-month, 72-month, 96-month, and 120-month survival rates of patients were 89.4%, 83.3%, 74.9%, 64.7%, 59.5%, and 59.5%, respectively, in patients with non-AKI; and 0%, 0%, 0%, 0%, 0%, and 0%, respectively, in patients with AKI stage 1; and 47.1%, 0%, 0%, 0%, 0%, and 0%, respectively, in patients with AKI stage 2 or 3. In a subgroup of serum albumin ≤3.5 g/dL, overall survival was significantly different according to AKI stage ($p<0.001$). The 12-month, 24-month, 48-month, 72-month, 96-month, and 120-month survival rates of patients were 88.1%, 80.6%, 58.4%, 37.0%, 37.0%, and 24.7%, respectively, in patients with non-AKI; and 66.7%, 0%, 0%, 0%, 0%, and 0%, respectively, in patients with AKI stage 1; and 0%, 0%, 0%, 0%, 0%, and 0%, respectively, in patients with AKI stage 2 or 3. Finally, we evaluated the effect of AKI stage on overall survival of HCC patients undergoing TACE according to presence of ascites. In a subgroup of patients without ascites, overall survival was significantly different according to AKI stage ($p<0.001$). The 12-month, 24-month, 48-month, 72-month, 96-month, and 120-month survival rates of patients were 89.9%, 83.7%, 72.2%, 60.2%, 56.0%, and 52.0%, respectively, in patients with non-AKI; and 50%, 0%, 0%, 0%, 0%, and 0%, respectively, in patients with AKI

stage 1; and 0%, 0%, 0%, 0%, 0%, and 0%, respectively, in patients with AKI stage 2 or 3. In a subgroup of patients with ascites, overall survival was significantly different according to AKI stage (*p*<0.001). The 12-month, 24-month, 48-month, 72-month, 96-month, and 120-month survival rates of patients were 81.7%, 72.6%, 57.2%, 22.9%, 22.9%, and 22.9%, respectively, in patients with non-AKI; and 0%, 0%, 0%, 0%, 0%, and 0%, respectively, in patients with AKI stage 1; and 0%, 0%, 0%, 0%, 0%, and 0%, respectively, in patients with AKI stage 2 or 3. Therefore, AKI stage is an independent risk factor for overall survival of HCC patients undergoing TACE regardless of BCLC tumor stage, serum albumin level and presence of ascites.

## Discussion

This study investigated the effect of AKI on the prognosis of HCC patients receiving TACE with Child-Pugh class A and pre-TACE SCr ≤1.5 mg/dL. AKI was evaluated using AKI stage 1–3 based on the ICA-AKI criteria. This study assessed if the updated AKI criteria predict overall survival of HCC patients treated with TACE. The AKI within four months after TACE developed in 11% of enrolled patients. The significant risk factors for overall survival after TACE were serum albumin level, bilobar tumor location, tumor stage, and AKI stage.

TACE is a nonsurgical intervention for the treatment of unresectable HCC. An image guided catheter is used to deliver chemotherapeutic agents (doxorubicin or cisplatin) and embolization materials into the supplying blood vessels for the tumor in the liver [16]. The patients treated with TACE are considered as a high risk group of AKI development. The incidence rate of AKI development after TACE was 9.1–23.8% [17–20]. Because the angiography using contrast media is conducted during TACE, contrast induced AKI develops after TACE. The mechanism of contrast associated AKI is as follow. Direct nephrotoxic effect of contrast on renal tubular epithelium is characterized by the loss of tubular cell polarity because of the redistribution of Na+/K+-ATPase from the basolateral to the luminal surface of the tubular cells [21]. These changes in renal tubular cells induces loss of cell function, apoptosis and necrosis. Also, tubular epithelial injury of contrast leads to renal vasoconstriction, luminal obstruction, increased intratubular pressure, and eventually, decreased in the glomerular filtration rate [21]. Renal dysfunction is one of common complications of liver cirrhosis, especially in patients with ascites [22]. Desai *et al.* reported that the prevalence of AKI in hospitalized cirrhotic patients had significantly increased in US [23]. This study showed that the presence of ascites was an independent risk factor for AKI in patients with HCC underwent TACE. Hayashi, *et al.* reported that the presence of ascites was significantly associated with an increase in SCr level in patients with HCC after TACE [24]. Hsu, *et al.* showed that AKI developed in 12.6% of HCC patients with ascites who underwent TACE and post-TACE AKI is a poor prognostic factor for overall survival [20]. Considering the findings of previous studies and this study, ascites is a significant risk factor for AKI development in patients with HCC who underwent TACE.

The concept of AKI has been updated to evaluate the early change of renal function and assess the severity of renal injury using a staging system [25–28]. Considering these concepts of AKI, the ICA-AKI criteria defined the diagnosis of AKI in liver cirrhosis as an elevation in SCr of ≥50% from baseline or an elevation in SCr ≥0.3mg/dl within 48 hours. The ICA-AKI criteria categorized the stage of AKI as stage 1–3 regarding the severity of renal injury [10]. This study was planned to assess the effect of AKI on the prognosis of HCC patients treated with TACE. We examined the risk factor for overall survival in HCC patients who underwent HCC. This study indicated that serum albumin level, tumor stage, bilobar location of tumor, and AKI stage significantly associated with overall survival in HCC patients who underwent TACE. Advanced tumor stage and bilobar location of tumor reflects the high burden of HCC

related to poor prognosis of patients. Serum albumin level means the underlying liver function of HCC patients.

This study was conducted to validate the updated AKI concept based on the ICA-AKI criteria as a predictor for the prognosis of HCC patients after TACE. The development of AKI within four months after TACE was 11% (37/347) in this study. The findings of this study showed that the AKI stage based on the ICA-AKI criteria predicted the prognosis of HCC patients who underwent TACE. Overall survival of the patients was poorer as the AKI stage increased. The ICA-AKI criteria predicts the prognosis of cirrhotic patients without HCC. The ICA-AKI criteria predicts the prognosis of cirrhotic patients without HCC. The ICA-AKI criteria predicts in-hospital mortality in cirrhotic patients treated for cellulitis [29]. Also, overall survival in cirrhotic patients with gastric variceal hemorrhage was poorer as the ICA-AKI stage increased [30]. AKI was an independent risk factor for overall survival in cirrhotic patients with spontaneous bacterial peritonitis. The ICA-AKI criteria was useful to predict the prognosis of cirrhotic patients with spontaneous bacterial peritonitis [31]. When the findings of the above studies were considered, the ICA-AKI criteria can facilitate predicting the prognosis of patients with liver cirrhosis including HCC.

This study has limitations. First, the study did not check other biomarkers for AKI other than SCr. Recent studies suggested the new biomarkers such as neutrophil gelatinase-associated lipocalin (NGAL), interleukin-18 (IL-18), kidney injury molecule 1 (KIM-1), liver-type fatty acid–binding protein (L-FABP), calprotectin, urinary angiotensinogen, and cystatin C [32]. Second, this study included a fair number of HCC patients with early stage: 147 patients (42%) in BCLC stage 0/A. Most of those cases were unsuitable for surgery or local ablation therapy. Finally, we could not evaluate the volume of contrast medium because the study was conducted in the retrospective manner using a historical cohort. Therefore, this study could not assess the effect of the volume of contrast medium on AKI development after TACE for HCC. Despite these limitations, this study demonstrates that the role of AKI on the prognosis of HCC patients who underwent TACE and validates the updated AKI concept based on the ICA-AKI criteria as a predictor for the prognosis of HCC treated with TACE. Also, we think the enrolled patients of this study could be considered representative of a larger population because the study was conducted based on the consecutive patients treated with TACE for HCC between January 2000 and December 2014.

## Conclusions

AKI is a crucial prognostic factor for mortality in patients with HCC undergoing TACE. The AKI stage and risk factors for AKI development facilitate evaluating the prognosis of HCC patients treated with TACE.

## Supporting information

**S1 File. Summary of clinical variables in the dataset.**
(DOCX)

## Author Contributions

**Conceptualization:** Won Sohn, Byung Ik Kim.

**Data curation:** Cheol Bae Ham, Nam Hee Kim.

**Formal analysis:** Won Sohn.

**Supervision:** Byung Ik Kim.

**Writing – original draft:** Won Sohn, Cheol Bae Ham.

**Writing – review & editing:** Nam Hee Kim, Hong Joo Kim, Yong Kyun Cho, Woo Kyu Jeon, Byung Ik Kim.

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
