## [Decision Letter · Decision Letter 0]

5 Oct 2020

PONE-D-20-22534

Effect of Acute Kidney Injury on the Patients with Hepatocellular Carcinoma Undergoing Transarterial Chemoembolization

PLOS ONE

Dear Dr. Kim,

Thank you for submitting your manuscript to PLOS ONE. After careful consideration, we feel that it has merit but does not fully meet PLOS ONE’s publication criteria as it currently stands. Therefore, we invite you to submit a revised version of the manuscript that addresses the points raised during the review process.

Two reviewers and I have pointed out that there are few issues related to statistical analysis and patients' characteristics.

We look forward to receiving your revised manuscript.

Kind regards,

Do Young Kim, MD, PhD

Academic Editor

PLOS ONE

Journal Requirements:

2. During our internal review, we noticed you have overlapping text with an abstract of your work published here: https://aasldpubs.onlinelibrary.wiley.com/doi/10.1002/hep.30941#

PLOS ONE cannot (re)publish material without sufficient permission from the original copyright holder to publish under a CC BY license. Please provide proof that the owner of the content (a) has given you written permission to use it, and (b) has approved of the CC BY license being applied to their content. You may have the following form completed by the owner as proof: https://journals.plos.org/plosone/s/file?id=7c09/content-permission-form.pdf. Alternatively, you may electronically request permissions electronically from the copyright owner and send us proof of approval, as long as the approval clearly shows that the owner has approved of the CC BY license being applied to their content. Please see https://journals.plos.org/plosone/s/licenses-and-copyright for more information

3. In your Methods section, please provide additional information about the participants and recruitment method and the demographic details of your participants. Please ensure you have provided sufficient details to replicate the analyses such as:

a) the recruitment date range (month and year),

b) a description of any inclusion/exclusion criteria that were applied to participant recruitment,

c) a statement as to whether your sample can be considered representative of a larger population,

d) a description of how participants were recruited, and

e) a reference for the historical cohort of Kangbuk Samsung Hospital

4. Please provide additional details regarding participant consent. In the ethics statement in the Methods and online submission information, please ensure that you have specified (1) whether consent was obtained, (2) whether consent was informed, and (3) what type you obtained (for instance, written or verbal, and if verbal, how it was documented and witnessed). If your study included minors, state whether you obtained consent from parents or guardians. If the need for consent was waived by the ethics committee, please include this information.

"NO"

6. Thank you for stating the following in your Competing Interests section: 

"No"

7. PLOS requires an ORCID iD for the corresponding author in Editorial Manager on papers submitted after December 6th, 2016. Please ensure that you have an ORCID iD and that it is validated in Editorial Manager. To do this, go to ‘Update my Information’ (in the upper left-hand corner of the main menu), and click on the Fetch/Validate link next to the ORCID field. This will take you to the ORCID site and allow you to create a new iD or authenticate a pre-existing iD in Editorial Manager. Please see the following video for instructions on linking an ORCID iD to your Editorial Manager account: https://www.youtube.com/watch?v=_xcclfuvtxQ

Reviewers' comments:

Reviewer's Responses to Questions

**Comments to the Author**

1. Is the manuscript technically sound, and do the data support the conclusions?

Reviewer #1: Yes

Reviewer #2: Yes

2. Has the statistical analysis been performed appropriately and rigorously? 

Reviewer #1: Yes

Reviewer #2: Yes

3. Have the authors made all data underlying the findings in their manuscript fully available?

Reviewer #1: Yes

Reviewer #2: Yes

4. Is the manuscript presented in an intelligible fashion and written in standard English?

Reviewer #1: Yes

Reviewer #2: Yes

5. Review Comments to the Author

Reviewer #1: The authors concluded that AKI is a crucial prognostic factor for overall survival in HCC patients undergoing TACE. Appropriately written article on a recently focused issue. However, this article had a couple of critical points to be addressed, as follows:

<major>

#1. BCLC stage was significantly associated with AKI development. It should be better to assess the multicollinearity among predictor variables in Cox regression model.

<minor>

#1. There is a typo in the sentence below (page 7, baseline characteristics section):

BCLC stages were non-AKI in 23 patients (6%), stage A in 124 patients (36%), stage B in 93 patients (27%), and stage C in 107 patients (31%).

#2. It is difficult to check the back part of Table 2. Please modify the Table 2.

#3. Please, upload modified figures with improved resolution.</minor></major>

Reviewer #2: Dr Sohn has studied the effect of AKI on HCC patients undergoing TACE. The findings are important and have a clinical impact. However I have several minor comments and requests before being accepted.

1. Please show the effect AKI on survival based on subgroup analysis according to 1) BCLC stage 0/A and BCLC BC, 2) albumin level with cut off 3.5 and 3) presence of ascites as shown in the results.

2. Have you ruled out the possibility of sepsis, the use of nephrotoxic drug or increasing ascites that could have been the cause for AKI?

3. The cisplatin based treatment is associated with occurrence of AKI compared to doxorubicin. However, have you observed the volume of contrast medium which could have affected AKI?

Minor comment: Final column of table 2 does not fit into page and was unreadable.

6. PLOS authors have the option to publish the peer review history of their article (what does this mean?). If published, this will include your full peer review and any attached files.

Reviewer #1: No

Reviewer #2: No

---

## [Author Response · Author response to Decision Letter 0]

1 Nov 2020

Dear editor and reviewers

Thank you for giving me the opportunity to revise our manuscript. Please find the enclosed revised version of the manuscript: PONE-D-20-22534 ‘Effect of Acute Kidney Injury on the Patients with Hepatocellular Carcinoma Undergoing Transarterial Chemoembolization’. We answered the editor’s and reviewers’ comments point by point and revised the manuscript to carefully address all the concerns raised by the handling editor and reviewers. We attached the file, 'Response to reviewers'. We thank to the editor and the reviewers for the informative reviews and now hope that the revised manuscript is now suitable for the publication in the “Plos One”.

With best wishes. 

Yours sincerely.

Byung Ik Kim, M.D., Ph.D.

---

## [Decision Letter · Decision Letter 1]

26 Nov 2020

Effect of Acute Kidney Injury on the Patients with Hepatocellular Carcinoma Undergoing Transarterial Chemoembolization

PONE-D-20-22534R1

Dear Dr. Kim,

We’re pleased to inform you that your manuscript has been judged scientifically suitable for publication and will be formally accepted for publication once it meets all outstanding technical requirements.

Kind regards,

Do Young Kim, MD, PhD

Academic Editor

PLOS ONE

Additional Editor Comments (optional):

Reviewers' comments:

Reviewer's Responses to Questions

**Comments to the Author**

1. If the authors have adequately addressed your comments raised in a previous round of review and you feel that this manuscript is now acceptable for publication, you may indicate that here to bypass the “Comments to the Author” section, enter your conflict of interest statement in the “Confidential to Editor” section, and submit your "Accept" recommendation.

Reviewer #1: All comments have been addressed

Reviewer #2: All comments have been addressed

2. Is the manuscript technically sound, and do the data support the conclusions?

Reviewer #1: Yes

Reviewer #2: Yes

3. Has the statistical analysis been performed appropriately and rigorously? 

Reviewer #1: Yes

Reviewer #2: Yes

4. Have the authors made all data underlying the findings in their manuscript fully available?

Reviewer #1: Yes

Reviewer #2: Yes

5. Is the manuscript presented in an intelligible fashion and written in standard English?

Reviewer #1: Yes

Reviewer #2: Yes

6. Review Comments to the Author

Reviewer #1: (No Response)

Reviewer #2: I appreciate the authors effort in presenting the response to the comments.

Since there is no significant difference according to the subgroup analysis, I suggest the authors to briefly mention that "Presence of AKI had no effect on survival in subgroup analysis according to BCLC stage, albumin level, and presence of ascites."

7. PLOS authors have the option to publish the peer review history of their article (what does this mean?). If published, this will include your full peer review and any attached files.

Reviewer #1: No

Reviewer #2: No

---

## [Editor Report · Acceptance letter]

2 Dec 2020

PONE-D-20-22534R1 

Effect of Acute Kidney Injury on the Patients with Hepatocellular Carcinoma Undergoing Transarterial Chemoembolization 

Dear Dr. Kim:

I'm pleased to inform you that your manuscript has been deemed suitable for publication in PLOS ONE. Congratulations! Your manuscript is now with our production department. 

Kind regards, 

on behalf of

Prof. Do Young Kim 

Academic Editor

PLOS ONE